# Dietary Habits and Psychological States during COVID-19 Home Isolation in Italian College Students: The Role of Physical Exercise

**DOI:** 10.3390/nu12123660

**Published:** 2020-11-28

**Authors:** Stefano Amatori, Sabrina Donati Zeppa, Antonio Preti, Marco Gervasi, Erica Gobbi, Fabio Ferrini, Marco B. L. Rocchi, Carlo Baldari, Fabrizio Perroni, Giovanni Piccoli, Vilberto Stocchi, Piero Sestili, Davide Sisti

**Affiliations:** 1Department of Biomolecular Sciences, University of Urbino Carlo Bo, 61029 Urbino, Italy; s.amatori1@campus.uniurb.it (S.A.); sabrina.zeppa@uniurb.it (S.D.Z.); marco.gervasi@uniurb.it (M.G.); erica.gobbi@uniurb.it (E.G.); f.ferrini2@campus.uniurb.it (F.F.); marco.rocchi@uniurb.it (M.B.L.R.); fabrizio.perroni@uniurb.it (F.P.); giovanni.piccoli@uniurb.it (G.P.); vilberto.stocchi@uniurb.it (V.S.); davide.sisti@uniurb.it (D.S.); 2Department of Neurosciences, University of Turin, 10124 Turin, Italy; antonio.preti@unito.it; 3Faculty of Psychology, eCampus University, 22060 Novedrate, Italy; carlo.baldari@uniecampus.it

**Keywords:** diet, exercise, lifestyle, mood, quarantine, young adults

## Abstract

Social isolation has adverse effects on mental health, physical exercise, and dietary habits. This longitudinal observational study aimed to investigate the effects of mood states and exercise on nutritional choices, on 176 college students (92 males, 84 females; 23 ± 4 years old) during the COVID-19 lockdown. During 21 days, nutrition and exercise were daily monitored, and the mood states assessed. A factor analysis was used to reduce the number of nutritional variables collected. The relationships between exercise, mood and nutrition were investigated using a multivariate general linear model and a mediation model. Seven factors were found, reflecting different nutritional choices. Exercise was positively associated with fruit, vegetables and fish consumption (*p* = 0.004). Depression and quality of life were, directly and inversely, associated with cereals, legumes (*p* = 0.005; *p* = 0.004) and low-fat meat intake (*p* = 0.040; *p* = 0.004). Exercise mediated the effect of mood states on fruit, vegetables and fish consumption, respectively, accounting for 4.2% and 1.8% of the total variance. Poorer mood states possibly led to unhealthy dietary habits, which can themselves be linked to negative mood levels. Exercise led to healthier nutritional choices, and mediating the effects of mood states, it might represent a key measure in uncommon situations, such as home-confinement.

## 1. Introduction

The novel coronavirus SARS-CoV-2, responsible for the COVID-19 epidemic, was first identified in Wuhan (China) in December 2019. In Italy, the first official cases appeared in late February, and on 9 March, the Italian government placed about 60 million people in a de facto quarantine mode, being asked to live in home-confinement for several weeks, until 4 May.

In the last months, several articles have been published regarding the effects of the home confinement and social-isolation period on different domains: psychological states [1,2], physical activity [3], nutrition [4,5] or the integration of these [6,7,8,9]. Mood disturbances, such as anxiety, depression, anger and irritability, have been previously reported in several studies during quarantine periods, as reviewed by Brooks et al. [10]. Ammar et al. [1] conducted an online survey during the COVID-19 outbreak on over a thousand people in home confinement, reporting reduced mental well-being and satisfaction and increased depression and need for psychological support, compared to the pre-epidemic period. The increased adverse psychological effects were associated with a reduction in physical activity and an increase in unhealthy diet behaviours [6].

### 1.1. Mood and Nutrition

The association between diet and mood has been recently systematically reviewed by Arab et al. [11]. The food intake can be involved in regulating mood and emotions, and this can affect the food choices suggesting a bidirectional influence [12]. Indeed, low mood and poor nutrition are mutually connected, one influencing the other and vice versa [13]. For example, it has been reported an alteration in food choices, not always healthy, in response to different psychological states or stress conditions. For example, foods rich in fat and carbohydrate are commonly preferred to be consumed by depressed subjects [14]. On the flip side of the coin, healthy dietary patterns (e.g., Mediterranean diet) are associated with a reduced risk of depression and better mental health [15]. For instance, an inverse association between vegetables and fruits consumption with depression mood has been previously found [16]. Furthermore, the intake of nuts containing unsaturated fatty acids, polyphenols and vitamins may have protective effects against mood and cognitive disorders [17,18]. Lassale et al. [15] have been able to synthesise the link between diet quality, measured using a range of predefined indices, and depressive outcomes, concluding that adherence to a healthy diet, in particular a traditional Mediterranean diet, or avoiding a pro-inflammatory diet, appears to confer some protection against depression in observational studies. This provides a reasonable evidence base to assess the role of dietary interventions to prevent depression [15].

### 1.2. Exercise and Nutrition

The phenomenon known as “multiple health behaviour change” defines the interrelation between health-related behaviours. This concept can be summarised as: the higher the probability of one health-related behaviour, the higher the probability for the other [19]. Indeed, physical exercise and diet were reported to be strictly interconnected [20]. Joo et al. [21] reported a modification in the dietary patterns of young adults after 15 weeks of training. They highlighted that, even if different effects were produced depending on the duration and intensity of the exercise, the overall trend was towards healthier food selection. Donati Zeppa et al. [22] reported that high-intensity exercise promoted a spontaneous and unaware modulation of food choices, leading to a healthier diet in young adults. Several mechanisms have been proposed to explain this phenomenon, as improved appetite control and a reduction in emotional overeating [23]; healthy nutritional choices and physical exercise seem to share the same pathways which lead to effective behavioural changes towards a healthier lifestyle [24].

### 1.3. Mood, Exercise and Nutrition

The interaction between exercise, improved psychological states and healthier eating behaviours was previously proposed by Annesi et al. [25], who reported a reduction in negative moods, an increase in self-regulate eating ability, and a clinically relevant reduction in weight, after six months of moderate-intensity exercise in obese women. Regular exercise practice was shown to positively influence mood, self-efficacy and use of self-regulatory skills, key predictors of controlled eating [26]. However, the direction of relationships between psychological and behavioural variables is difficult to assess. Annesi and Mareno [25,27] used a mediation model to study these interactions, suggesting a psychological pathway between increased exercise and decreased emotional eating, so highlighting the role of exercise in facilitating nutrition changes.

### 1.4. The COVID-19 Context

Given the above, the effects of unique conditions—as it has been the COVID-19 outbreak—on psychological states, dietary behaviours and physical activity have been a topic of great interest. Antunes et al. [7] conducted a survey on a sample of Portuguese adults during the epidemic, reporting altered eating habits, with increased quantity and frequency of meals; contrasting results are presented regarding the attention to the food selection. Di Renzo et al. [28] investigated the impact of the COVID-19 pandemic on eating habits and lifestyle changes among the Italian population aged ≥12 years, reporting a higher adherence to the Mediterranean diet in the group aged 18–30 years, compared to the younger and the elderly population. Gallo et al. [8] investigated the impact of isolation measures on energy intake and physical activity levels in Australian university students, reporting a reduction in physical exercise levels, and an increase in energy intake and snacking frequency. Interestingly, Romero-Blanco et al. [29] reported that—during home-confinement in Spain—university students that adopted a Mediterranean diet, also spent more time doing physical exercise. Ingram et al. [30] investigated the changes in health behaviours related to negative mood states in a sample of about four hundred adults during the COVID-19 quarantine in the UK, reporting changes in diet, sleep quality, physical activity levels and poorer mental health. Most of the studies above reported had cross-sectional designs, performed through questionnaires or online surveys. It was previously identified that surveys and self-reported food diaries with a short time of data collection are likely to underreporting from participants; however, the 24 h recall for multiple days significantly reduces this effect [22].

The present longitudinal observational study aimed to investigate the effects of mood states and exercise on nutritional choices of a sample of college students, during the period of forced home-isolation due to COVID-19 outbreak. A daily follow-up for three weeks has been done, recording both dietary habits and physical exercise of the participants.

## 2. Materials and Methods

### 2.1. Participants

A convenience sampling was used in this study. Participants were college students, pertaining to three different study courses (pharmacy, biology and sports sciences) at the Urbino University, following online classes during the data collection period. During the lockdown period, people were allowed to leave home only for health issues, to go to work or to buy food and other essential products (only one family member at the time): therefore, people’s movements were limited to the minimum necessary. Bars and restaurants were closed, so people were forced to eat at home. They were invited to take part in the study through verbal explanation of the project during an online class, and later, an email was sent to each of them. All participants read and signed a written informed consent to take part in the study, which was conducted in accordance with the Helsinki Declaration. They were free to leave the study at any time, without giving any further justification. Once data was collected by the researcher, names were removed and an univocal identity code assigned to every participant, to secure anonymity. Ethics committee approval has been received by the Human Research Ethical Committee of the Urbino University (No. 31_2020).

### 2.2. Data Collection

Data collection was conducted during three weeks of home-isolation due to the COVID-19 pandemic in Italy, between 6–26 April 2020 (Figure 1). Before starting to record the diet, participants were asked to add in a pre-structured form their height and weight, if they were spending the isolation period at home with their families, with friends or alone, and if they were used to regularly practicing sports activity before the COVID-19 epidemics. Exclusion criteria were being on a particular dietary regimen (e.g., prescribed by a nutritionist), suffering from chronic diseases, being afflicted by COVID-19 before or during the period, and having a flu during the data collection period.

### 2.3. Dietary Habits

Participants were asked to daily fill a pre-structured food diary, in which they inserted each food or drink consumed during the whole 24 h, with the quantity (expressed in pieces, e.g., one banana, or grams, e.g., 100 g of pasta). A daily reminder to fill the food diary was sent to every participant during the study period. Diet monitoring has gone on for 21 days (three weeks). After the completion of the data collection, data were manually screened and uploaded on the MètaDieta software (METEDA Srl, San Benedetto del Tronto, Italy), used for the data processing. This software was previously used for monitoring the diet in university students [22]. It automatically calculates the total energy intake and the nutritional values, with macro- and micronutrients quantities, for each food or drink consumed during the day. The following variables were selected for the analyses: carbohydrates (glucides, starch, glycaemic index, glycaemic load), protein (animal-based protein, plant-based protein, protein value), lipids (animal lipids, plant lipids, mono-, polyunsaturated and saturated fats, Omega 3 and 6), fibre (soluble and insoluble), amino acids (tryptophan, phenylalanine, tyrosine) and vitamins (A, B6, B12, C, D, E, and folic acid). Furthermore, the distribution of energy intake among the different meals throughout the day has been evaluated.

### 2.4. Physical Exercise

On a separate sheet of the food diary, participants were instructed to add if they trained, with the duration in minutes and the perceived effort (RPE) using the modified CR-10 scale by Foster et al. [31], for each day. The CR-10 scale asked participants to rate their effort answering the simple question “How was your workout?” on a scale from 0 “rest” to 10 “maximal”. The participants were already familiarised with the use of this scale; furthermore, verbal and written instructions on the use of the CR-10 scale and the diary have been provided to all the participants before the beginning of the study. Duration and intensity values were used to calculate the training load, as SessionRPE (duration × intensity) [32]. This method was reported to be superior to monitor training loads of fitness workout sessions compared to other heart-rate based methods [33]. The training load has been used as the measure of exercise performed for the statistical analyses. Due to the home isolation restrictions, people were not allowed to go out to do physical exercise; assuming that other forms of physical activities were minimal in this situation, only structured exercise sessions (e.g., runs or rides on treadmill or bike trainers, free-weight exercises) were taken into account.

### 2.5. Psychological Scales

#### 2.5.1. Positive and Negative Affect Schedule (PANAS)

The PANAS is a tool that assigns scores based on feeling or emotion-related adjectives [34]. Wordings are rated as positive when it refers to positive emotions and positive interactions with others (e.g., interested, strong, proud). Wordings are rated negative when they involve negative experiences with the world and others (e.g., hostile, nervous, afraid). The brief 20-item scale was used in this study. On each adjective, respondents assign a score on a Likert-type scale ranging from 1 (very slightly) to 5 (extremely). This tool was previously used with a university student population [35]. The Italian validated version of the PANAS was used in this study [36].

#### 2.5.2. Patient Health Questionnaire 9 (PHQ-9)

The PHQ-9 is a self-administered questionnaire aimed at surveying the nine main criteria for major depression [37,38]. Participants have to rate, on a 4-point scale (from 0 “not at all” to 3 “nearly every day”), how often they have been bothered by the nine criteria over the last two weeks (e.g., “Little interest or pleasure in doing things”, “Feeling down, depressed, or hopeless”, “Feeling tired or having little energy”). A global score was calculated as a sum of the scores on each item. This questionnaire was validated for depression screening in university students [39]. For this study, we used the validated Italian version of the scale [40]. An oft-used scoring threshold for the presence of depression is: 0–4 = none, 5–9 = mild, 10–14 = moderate, 15–19 = moderately severe, ≥20 = severe.

#### 2.5.3. 12-Item Short Form Health Survey (SF-12)

The SF-12 is a self-report measure of the impact of health conditions on a person’s everyday life [41]. It is used as a measure of health-related quality of life (HR-QoL), previously used with university students [42]. The tool covers eight main domains, and these domains can be grouped into two main areas: mental-depending HR-QoL (e.g., “How much of the time during the past 4 weeks have you felt calm and peaceful?”) and physical-depending HR-QoL (“During the past 4 weeks have you accomplished less than you would like with your work or other regular activities as a result of your physical health?”). A global score of HR-QoL can also be calculated by summing the scores on each item. Higher scores on the SF-12 correspond to higher HR-QoL. In this study, we used the validated Italian version of the SF-12 [43].

### 2.6. Statistical Analyses

Descriptive statistics were reported for macronutrients and demographics characteristics. Mean (standard deviation) was used for demographic variables; median (first (Q1) and third (Q3) quartiles) for training frequencies (sessions/week), psychometric scales’ scores and percentage of energy intake distributed in the different meals throughout the day (breakfast, morning snack, lunch, afternoon snack and dinner). From the list of macro- and micronutrients obtained from the Metadieta software, given the high number of variables collected, it was appropriate to reduce their number, minimising the loss of information. So, a Principal Axis Factor with a Varimax (orthogonal) rotation of 27 nutritional variables above considered was conducted on data gathered from participants. Kaiser–Meyer Olkin (KMO) measure of sampling adequacy was also performed; the minimum acceptable value for KMO is 0.6, although the ideal is over 0.70. Only factors with an eigenvalue of ≥1 were considered. The variance percentage accounted for by each component to the total variance was also reported. The factor score coefficient matrix was also computed. The resulting component score variables are representative of—and can be used in place of—original variables with an acceptable loss of information (1-total cumulative variance).

The internal consistency reliability index for the mood states questionnaire was quantified using Cronbach’s alpha. Cronbach’s alpha is a measure of internal consistency, that is how closely related a set of items are as a group. It is considered ‘moderate’ when it is >0.6 and ‘good’ when it is >0.7 [44].

In order to investigate whether psychological states and exercise could be associated with the nutritional choices, a multivariate generalised linear model was performed. The dependent variables were the factors obtained from factor analysis; participant’s repeated measures (ID) was a random factor; the fixed factors was sex; finally, training load, mood scales (PANAS, PHQ-9, SF-12) and BMI were covariates. Pillai’s trace statistic was used; Pillai’s trace is more robust to departures from assumptions than Wilk’s lambda, Hotelling’s trace, and Roy’s largest root. Subsequent univariate ANOVAs were also performed for each dependent variable (factors) considered.

Finally, in order to assess if exercise could mediate the effect of psychological states on nutrition, a mediation effect has been calculated. A mediated effect is also called ‘indirect effect’, and it occurs when the effect of the independent variables (psychometric scales) on the dependent variable (factors’ scores obtained by factor analysis) is—as the name says—mediated by another variable: a mediator (training load). Mandatory condition to establish any mediation effect is that the independent variable (IV) must have a significant effect on the mediator. In mediation analysis, we considered as independent variables the same variables reported in GLM analysis. In mediation analysis, the following estimations are reported:Average causal mediation effect: it is the indirect effect of the IV (psychological states) on the DV (factors scores) that goes through the mediator (training load).Average direct effect: it describes the direct effect of the IV on the DV, without mediator effect.Total effect: it stands for the total effect (direct + indirect) of the IV on the DV.Proportion Mediated: it describes the proportion of the effect of the IV on the DV that goes through the mediator. It is calculated by dividing the mediation effect through the total effect.

All elaborations were conducted with α = 0.05. Statistical analyses were performed using SPSS 22.0 (IBM, Armonk, NY, USA), except from the mediation effect that has been calculated with the *mediation* v.4.5.0 package [45] using R Studio v. 1.2.5033 software. GraphPad Prism 8 (GraphPad Software, San Diego, CA, USA) has been used to build the figures.

## 3. Results

### 3.1. Sample Characteristics

A total of 250 students were invited to participate in the research. Of these, 176 (response rate: 70.4%) participants (23 ± 4 years old) voluntarily took part in this observational study; all the participants correctly fulfilled the three week daily food diary. Of these 92 were males (178.5 ± 6.3 cm, 73.1 ± 9.8 kg, 22.9 ± 2.6 kg/m^2^) and 84 were females (164.6 ± 6.6 cm, 58.2 ± 7.5 kg, 21.4 ± 2.5 kg/m^2^). 141 of them reported practising sports activity, at different levels (from beginner to professional), with a median of four training sessions/week (Q1: 3, Q3: 5). Almost all of them (except seven people) reported having spent the whole lockdown period in their respective residences, with their families.

### 3.2. Dietary Habits

Energy intake and macro- and micronutrients were analysed daily. Average energy intake of 1742 (±688) kcal was reported in males, and 1237 (±486) kcal in females. Macronutrients were distributed in the diet as 48% carbohydrates, 30% lipids and 22% proteins. Energy intake was spread out during the day as so: (median, Q1–Q3): breakfast 14.0% (5.8–22.0%), morning snack 0.0% (0.0–2.0%), lunch 38.0% (29.0–50.0%), afternoon snack 6.0% (0.0–13.0%) and dinner 33.0% (22.0–44.0%). In Figure 2 are reported the average macronutrients (in absolute value, gr) and energy intake (kcal), split for gender. A complete description of the micronutrients can be found as Appendix A.

### 3.3. Factor Analysis

A Principal axis factor analysis with varimax rotation has been conducted to assess the underlying structure in micro and macronutrients. The analysis yielded a seven-factor solution; the Kaiser–Meyer Olkin measure of sampling adequacy was 0.783, above the minimum ideal value. Table 1 presents the rotated component matrix, eigenvalues, and the percentage of variance explained; note that the seven components explain 77.1% of the variance.

Seven variables loaded in Factor 1, accounting for 32.0% of the variance; a high intake of carbohydrates characterised it with elevated glycaemic load (including both glucides and starch), plant-based protein and fibre. This factor may be mainly due to cereals, and secondly to legumes consumption. Six variables loaded in Factor 2, accounting for 14.6% of the variance; these were related to vegetable-fats consumption (monounsaturated, polyunsaturated, and saturated fats), vitamin E and Omega-6. This factor might be interpreted as dried fruits and vegetable oils intake, as olive oil. Nine variables loaded in Factor 3, accounting for 9.3% of the variance; it was characterised by a high intake of animal-based protein and lipids, with high protein value, saturated and monounsaturated fats, Omega-6, vitamin B6 and Tyrosine, but showed a negative load for glucides consumption. This Factor could be summarised by milk, its derivatives and high-fat meats. Nine variables loaded in Factor 4, accounting for 7.4% of the variance; it was characterised by a prevalence of hydrosoluble (B6, C and folic acid) and liposoluble vitamins (A, E), fibres and plant-based protein with a high protein value. These items lead to fresh vegetables and fruit consumption (in particular wheat germs). Four variables loaded in Factor 5, accounting for 5.0% of the variance; it was characterised by a significant amino acid intake (tryptophan, phenylalanine, tyrosine) and animal-based proteins. Given that this factor is not affected by fats, it could be hypothesised that lean (low-fat) meats represent it. Five variables loaded in Factor 6, accounting for 4.6% of the variance; with a high intake of Omega-3, vitamin B12, vitamin D, animal-based proteins, and polyunsaturated fats, this factor could be easily interpreted as fish consumption. Finally, four variables loaded in Factor 7, accounting for 4.2% of the variance; it was characterised by soluble and insoluble fibres, with negative weights for glycaemic index and load. This Factor could be explained with whole-grain cereals consumption and other vegetables rich in fibres. The above-described factors are graphically summarised in Figure 3.

### 3.4. Physical Exercise

The exercise was daily monitored by calculating the training load as SessionRPE [32]. On average, participants trained 4.6 ± 3.3 times/week; the mean duration was 54 ± 41 min, with an RPE of 6.6 ± 1.8. As people were not allowed to go outside for training, exercise sessions mainly included indoor walks, runs or rides (on treadmill or bike trainers), rope jumping and free-weight exercises.

### 3.5. Psychological States

A total of 81 (88%) males and 78 (92.9%) females fully responded to the questionnaires.

The PHQ-9 scores had a median of 6 (Q1: 3, Q3: 8) for males and 7 (Q1: 6, Q3: 10) for females. A total of 28 males (30.4%) and 9 females (10.7%) were classified as ‘none symptoms’. A total of 41 (44.6%) and 49 (58.3%) participants had mild symptoms. A total of 11 (11.9%) and 16 (16.1%) reported moderate symptoms. A total of 1 (1.1%) male and 3 (3.6%) females reported moderately severe symptoms, and only 1 female (1.2%) had severe symptoms of depression. Median for the PANAS positive affect score was 31 (Q1: 26, Q3: 37) for males and 27 (Q1: 23.75, Q3: 34) for females, while for the PANAS negative affect score median was 18 (Q1: 15, Q3: 23) for males and 23 (Q1: 18.75, Q3: 29) for females. The SF-12 score was calculated as the total score. The median for the SF-12 score was 39 (Q1: 35, Q3: 41) for males and 37 (Q1: 34, Q3: 40) for females.

The alpha for PHQ-9 was 0.69 (reasonable internal consistency reliability), whilst alpha for PANAS was 0.63 (minimally adequate internal consistency reliability); finally alpha was 0.78 (good internal consistency reliability) for SF-12.

### 3.6. Effects of Mood and Exercise on Nutrition

A multivariate analysis of variance was conducted to assess if gender, BMI, training load and mood states were significantly associated with the linear combination of seven latent factors deriving from factor analysis. No different effects of training volume (duration) or intensity (perceived exertion) have been found, so training load was considered as the only measure of exercise performed. Box’s Test of Equality of Covariance Matrices showed a *p* < 0.001, so Pillai’s trace statistic was used. A significant difference was found for all predictor variables: gender, with Pillai trace = 0.019, F_(2593,7)_ = 7.02, *p* < 0.001; ID, with Pillai trace = 1.474, F_(18193,861)_ = 5.63, *p* < 0.001; PANAS positive score, with Pillai trace = 0.008, F_(2593,7)_ = 2.69, *p* = 0.009; PANAS negative score, with Pillai trace = 0.006, F_(2593,7)_ = 2.14, *p* = 0.037; SF12, with Pillai trace = 0.007, F_(2593,7)_ = 2.69, *p* = 0.009; PHQ-9 total score, with Pillai trace = 0.009, F_(2593,7)_ = 3.26, *p* = 0.002; BMI, with Pillai trace = 0.008, F_(2593,7)_ = 3.05, *p* = 0.003 and finally training load, with Pillai trace = 0.008, F_(2593,7)_ = 2.94, *p* = 0.004. In subsequent ANOVA analysis, ID was always significant for all dependent variables; this implies that the nutrition pattern is a personal characteristic with a large effect size (eta squared = 0.211).

Follow up univariate ANOVAs indicated that BMI (t = 1.97; *p* = 0.049), PHQ-9 (t = 2.83; *p* = 0.005) and PANAS positive (t = 3.01; *p* = 0.003) were positively correlated to Factor 1, whilst SF-12 (t = −2.88; *p* = 0.004) and gender (male = 0, female = 1; t = −4.96; *p* < 0.001) was negatively correlated. Factor 2 was associated only with ID, as above reported. Considering Factor 3, gender (t = −2.61; *p* = 0.009), PANAS negative (t = −1.99; *p* = 0.046) and training load (t = −2.24; *p* = 0.025) showed negative correlations whilst BMI (t = 2.18; *p* = 0.030) showed positive one. Factor 4 was positively correlated only to training load (t = 2.88; *p* = 0.004). BMI (t = 2.48; *p* = 0.013), PHQ-9 (t = 2.06; *p* = 0.040) and PANAS positive (t = 2.11; *p* = 0.035) were positively correlated to Factor 5, whilst SF-12 (t = −1.98; *p* = 0.048) and gender (t = −2.11; *p* = 0.035) were negatively correlated. Factor 6 was positively correlated to PANAS negative (t = 2.57; *p* = 0.010) and training load (t = 2.13; *p* = 0.033) and negatively correlated to PHQ-9 (t = −2.31; *p* = 0.021). Finally, as Factor 2, Factor 7 was associated only with ID. These results are graphically presented in Figure 4.

### 3.7. Mediation Effect

The mediation effect of exercise on psychometric scales has been investigated. As presented in the hypothetical model in Figure 5, we assumed that the influence of mood (psychometric scales) on the nutritional choices (factors above presented), might be mediated by the training load.

This analysis has been carried out with the *mediation* R package, with training load as mediating factor, psychometric scales (PHQ-9, PANAS +, PANAS − and SF-12 scores) as independent variables, and the seven factors as dependent variables. Firstly, the mediator has been regressed onto the independent variables; the training load was significantly correlated to each psychological scale (*p* < 0.05). The effect of mood (measured through the psychometric scales) on nutritional choices (factors’ scores) was mediated via the exercise performed, only for Factors 4 (fresh fruit and vegetable) and 6 (fish), as illustrated in Figure 6. For Factor 4, the mediation effect was −0.0014 (95% CI: −0.0029–0.00; *p* = 0.024) for PHQ-9 score, 0.0012 (95% CI: 0.0004–0.00; *p* < 0.001) for PANAS positive score, 0.0008 (95% CI: 0.0002–0.00; *p* = 0.004) for PANAS negative score, and −0.0017 (95% CI: −0.0031–0.00; *p* < 0.001) for SF-12 score. The proportion of the effect of the mood on Factor 4 that goes through the exercise was 2.7% for PHQ-9 (*p* = 0.024), 9.0% for PANAS positive (*p* < 0.001), and 3.8% for PANAS negative (*p* = 0.004) scores respectively. The model accounted for 4.2% of the variance in Factor 4 (R^2^: 0.0417, *p* < 0.001). For Factor 6, the mediation effect was −0.0018 (95% CI: −0.0039–0.00; *p* = 0.03) for PHQ-9 score, 0.0016 (95% CI: 0.0008–0.00; *p* < 0.001) for PANAS positive score, 0.0011 (95% CI: 0.0002–0.00; *p* = 0.008) for PANAS negative score, and −0.0022 (95% CI: −0.0039–0.00; *p* < 0.001) for SF-12 score. This means that the proportion of the effect of the PANAS positive scores on Factor 6 that goes through the exercise was 22.2% (*p* = 0.044). The model accounted for 1.8% of the variance in Factor 6 (R^2^: 0.0184, *p* < 0.001).

## 4. Discussion

This longitudinal observational study aimed to investigate the effects of mood states and exercise on nutritional choices during the COVID-19 home-confinement period. Based on the previous literature, it is known that both mood and exercise might affect nutritional choices [13,22]. We assumed that these effects are not independent, but exercise might mediate the mood-food relationship, so determining changes in nutritional choices [27].

### 4.1. Effect of Exercise on Nutrition

The combination of a healthy diet and physical exercise is the foundation of a healthy lifestyle. Exercise contributes to a negative energy balance by increasing energy expenditure, but the mechanisms which induce modifications in the dietary habits have not yet been completely elucidated. Indeed, research suggests that the role of physical exercise in stimulating psychosocial changes, which leads to better eating, might be more important than the effect of increased energy expenditure [27]. It has been proposed that exercise could influence nutrition habits by application of self-regulatory conducts [23], improved mood states and increased self-efficacy [25], being suggested as a gateway behaviour for healthful eating [46]. In our study, physical exercise has been shown to negatively correlate with Factor 3 and positively correlates to Factor 4 and 6. This means that in people who exercise more, have a higher consumption of fresh fruit, vegetable and fish, and a lower intake of fat-rich meats and milk-derivatives, the results suggest that physical exercise appears to mediate healthier dietary habits. This is in accordance with the results obtained by Donati Zeppa et al. [22], which showed that nine weeks of high-intensity interval training promoted a spontaneous modulation of healthy food choices and regulation of dietary intake in young normal-weight college students, aged 21–24. Jayawardene et al. [47] also highlighted that exercise could favour fruit and vegetable consumption in young adulthood. Romero-Blanco et al. [29] conducted a study among university students during home-confinement due to COVID-19 in Spain, and evidenced a positive correlation between a Mediterranean diet (known to be rich in fruit and vegetables) and physical exercise. Physical exercise and healthy food choices seem to facilitate each other, via improved self-regulatory strategies and intentions (e.g., planning to perform physical exercise), indicating potential transfer effects between these behaviours [19,47,48]. Although exercise has been observed to mainly influence fruit and vegetable consumption [27], it could be argued that the above-mentioned transfer effects may also influence other nutritional choices, such as increasing fish intake or limiting meats or milk-derivatives, known to be rich in saturated fats.

### 4.2. Effect of Mood on Nutrition

In our sample, 13% of males and 19% of females reported moderate to severe symptoms of depression (PHQ-9 ≥ 10), a result that is in accordance with the known higher prevalence of depression in women than in men [49,50]. Quarantine situations have been evidenced to negatively impact on mental health, in particular by promoting insomnia, depression, anxiety, irritability, stress-related disorders and anger [51,52]. Poorer mental health, a partial consequence of COVID-19 lockdown, possibly led to less healthy diets, which can themselves be linked to negative mood levels, such as anxiety and depression [53], in a vicious circle [30]. Indeed, associations between diet and psychological disorders—such as depression and anxiety—have been previously reported [54].

Higher scores on PHQ-9 (i.e., higher depression scores) are linked to a high intake of Factor 1 (mainly characterised by cereals and legumes) and Factor 5 (e.g., low-fat meats), and low consumption of Factor 6 (e.g., fish) consumption. The results of PHQ-9 scores are coherent with those reported for SF-12 (i.e., better quality of life), where higher scores were associated with lower consumption of Factor 1 and 5. There is a known link between mood and carbohydrates, with many people eating palatable high carbohydrate/high-fat foods when in a poor mood [55]. Indeed, carbohydrate-rich foods could be a way of self-medicating depression [56], and the effect of carbohydrate cravings may be even stronger when people eat food with a high glycaemic index. A carbohydrate-rich meal tends to raise serotonin levels, which imbalance contributes to the development of depressive symptoms [57]. Therefore, it would make sense that people with more severe depressive symptoms (likely associated with a lower perceived life quality) tend to have a diet rich in carbohydrates. Conversely, there is some, although controversial, evidence that eating fish might reduce the risk of depression [58,59]. In general, the more enriched in fruits and vegetables and poorer in fat and in red meat a diet is, the lower the chance of showing symptoms of depression [58]. However, people perceiving their quality of life as already good were less likely to consume foods popularly considered “healthy” since, in all likelihood, they did not feel the need of correcting with the diet their health or psychological status.

Furthermore, higher scores in the positive affect scale (PANAS positive) were associated with a higher intake of Factor 1 and 5. This could be explained by the model proposed by Christensen L. [60]: it has been proposed that there is a reciprocal interactive relationship between carbohydrate cravings and mood. Emotional distress causes a preference and craving for carbohydrate-rich foods, and these may have a positive reinforcing effect on mood, explaining the higher PANAS positive score. However, legumes consumption has been previously related to lower stress levels and better positive affects [61]. Finally, although negative affect (PANAS negative scores) have been associated with a reduction in Factor 3 (i.e., high-fat meat and milk-derivatives) and an increase in Factor 6 (i.e., fish), research relating meat consumption with mood and life quality is inconclusive, with contrasting results; so, more research is needed to investigate these relationships [61]. However, it might be suggested that this effect depends on a subgroup of people with modest depression preferring a diet with fish and less high-fat meat and milk-derivatives as a strategy to deal with their symptoms.

Diet and mood can mutually influence each other through the gut–brain axis, in which the gut microbiome plays an important role in maintaining mental health [62]. Brain areas and neurotransmitters and neuropeptides that are involved in both mood and appetite likely play a role in mediating this relationship [63].

### 4.3. Exercise as a Mediator between Mood and Nutrition

The indirect effects of exercise on mood scales (PHQ-9, PANAS positive and negative, SF-12) and nutritional choices (factors) were studied using a mediation model. Although some other studies reported that changes in diet and physical exercise had a clear link to negative mood states [30], to the authors’ knowledge only a few studies used a mediation approach to investigate this relationship [25,27]. Results supported that exercise partially mediated the relationship between the four psychological domains (PHQ-9, PANAS positive and negative, SF-12) and the Factor 4 (fruit and vegetable consumption) and 6 (fish consumption). This is in accordance with the study published by Annesi et al. [27], who claimed that changes in physical exercise mediated the relationships between mood and increased fruit and vegetable consumption. Mediation may still exist in the absence of a significant association between dependent and independent variables, as reported by MacKinnon et al. [64]. Indeed, in our case, none of the psychological scales was associated with Factor 4, and only PHQ-9 and PANAS negative were associated with Factor 6, respectively, negatively and positively (Figure 4). MacKinnon et al. [64] and Kenny [65] described inconsistent mediation when the mediator acts as suppressor variable, in one of the following cases: indirect effect and total effect are opposite in sign, the estimate of the total effect is closer to zero than the direct effect, or the direct effect is larger than the total effect [66]. Given the above, exercise could be a suppressor variable that buffers the interaction between SF-12 and Factor 4 and 6, PHQ-9 and PANAS negative and Factor 6. The mediation effects of exercise in the links of PHQ-9 and SF-12 on nutritional Factors might suggest that some participants actively changed their lifestyle to deal with the health and psychological anticipated impact of the lockdown and tried to prevent it by increasing their physical exercise and modifying their diets.

This work has some strengths and limitations. The main strength point is that diet monitoring was conducted for three-weeks using a 24 h recall method, instead of using simple surveys. Moreover, to our best knowledge, this is one of the first attempts to investigate the complicated relationship between dietary habits, psychological states and physical exercise using a mediation model. A limitation of the paper is that psychometric scales were collected only once during the monitoring period. Furthermore, we do not have data regarding the dietary habits pre-quarantine, which could have been useful in order to assess changes in nutrition and lifestyle behaviours consequent to the home-isolation. However, some effects of the Italian lockdown might reveal later than the short duration of this trial was able to observe. Another limitation is represented by the convenience sampling method, used for the recruitment of the participants. Future studies could focus on the psychological and physiological mechanisms underlying the changes in dietary habits following different training modalities.

## 5. Conclusions

In this paper, the analysis of nutritional choices, physical exercise and mood in college students during COVID-19 outbreak highlighted the importance of physical exercise on dietary habits. Indeed, exercise showed to have both a direct effect, influencing fresh fruit and vegetables and fish consumption, and an indirect effect on nutritional choices, counterbalancing the impact of negative psychological states on the dietary habits. Poorer mental health possibly led to unhealthy diets, which can themselves be linked to negative mood levels, in a vicious circle. Overall, exercise led to healthier nutritional choices and, mediating the effects of mood states; it might represent a key measure in particular situations, such as a home-confinement due to a pandemic.

## Figures and Tables

**Figure 1 nutrients-12-03660-f001:**
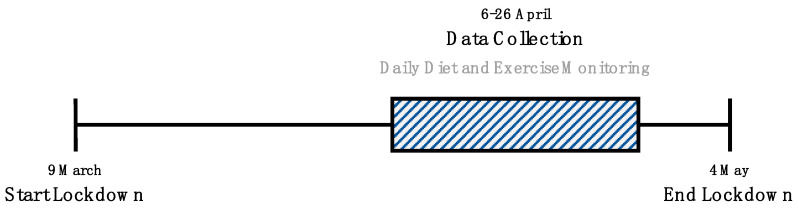
Study timeframe.

**Figure 2 nutrients-12-03660-f002:**
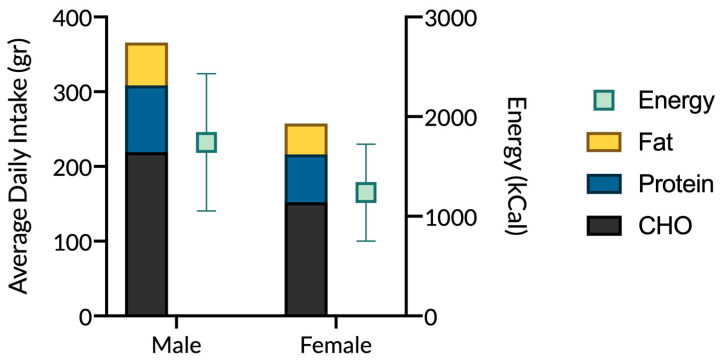
Daily macronutrients (gr) and energy (kcal) intake, for male and female.

**Figure 3 nutrients-12-03660-f003:**
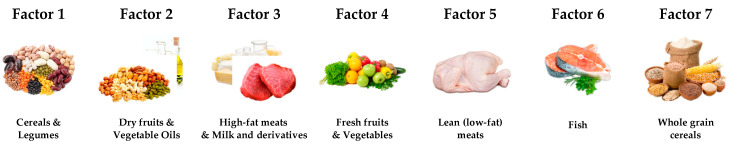
Factors possible explanations.

**Figure 4 nutrients-12-03660-f004:**
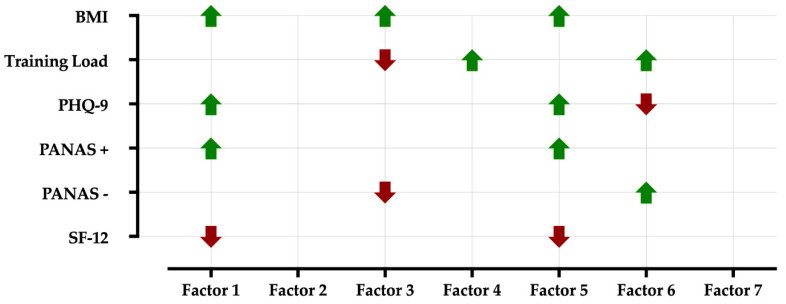
Interactions between Factors (dependent variables) and Body Mass Index (BMI), training load, and psychometric scales (PHQ-9, PANAS +, PANAS −, and SF-12). Green arrows show positive correlations, while red arrows negative correlations. Only significant interactions have been reported (*p* < 0.05).

**Figure 5 nutrients-12-03660-f005:**
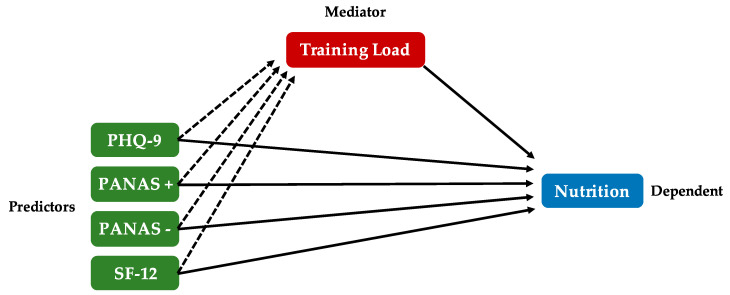
The theoretical model of the mediating effect of training load on the relationship between psychological states and nutritional choices. Solid lines represent a direct effect (predictors on dependent, mediator on dependent), while dashed lines represent the mediation effect.

**Figure 6 nutrients-12-03660-f006:**
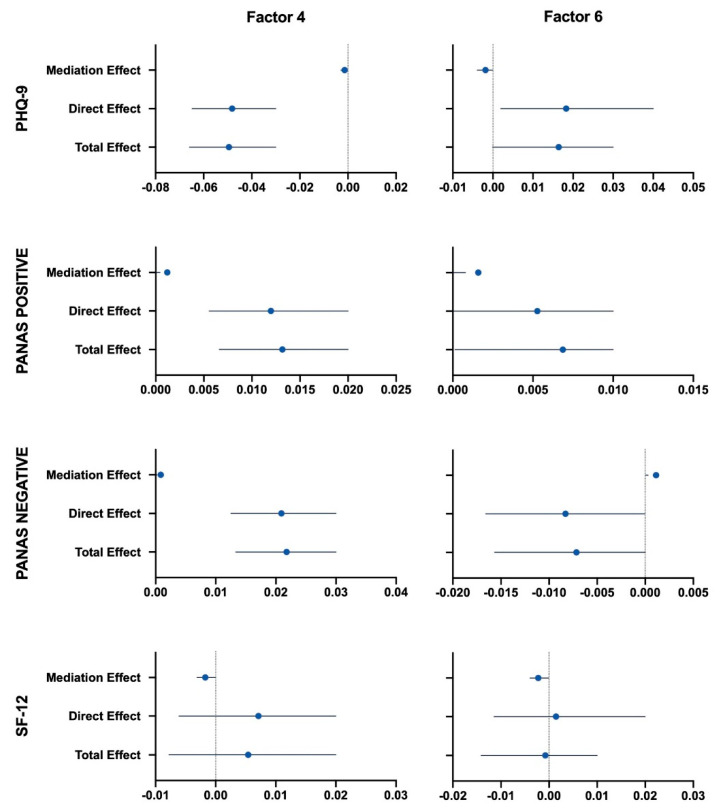
Plots of mediation, direct and total effects of training load on psychometric scales (PHQ-9, PANAS +, PANAS −, SF-12) for Factors 4 (fruit and fresh vegetables) and 6 (fish).

**Table 1 nutrients-12-03660-t001:** Factor analysis results. Only the components’ values significantly related to each factor are reported.

	Factors						
Component	1	2	3	4	5	6	7
Starch	0.883						
Carbohydrates	0.832						
Glycaemic load	0.821						−0.361
Plant-based protein	0.751			0.373			
Glucides	0.656		−0.450				
Vegetable fats		0.826					
Omega-6		0.747	0.362				
Monounsaturated fats		0.738	0.526				
Polyunsaturated fats		0.714				0.378	
Vitamin E		0.620		0.597			
Animal lipids			0.842				
Saturated fats		0.366	0.795				
Animal-based protein			0.659		0.483	0.378	
Protein value			0.559	0.377			
Vitamin C				0.767			
Folic acid				0.730			
Vitamin A				0.568			
Vitamin B6			0.357	0.555			
Insoluble fibre	0.368			0.549			0.520
Tryptophan					0.913		
Phenylalanine					0.903		
Tyrosine			0.395		0.830		
Vitamin D						0.864	
Omega-3						0.798	
Vitamin B12						0.537	
Glycaemic Index							−0.791
Soluble fibre	0.435			0.518			0.541
Eigenvalues	8.64	3.94	2.52	1.97	1.36	1.23	1.13
% of cumulative variance	32.0	46.6	55.9	63.3	68.3	72.9	77.1

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
