# Peer review of "Dietary Habits and Psychological States during COVID-19 Home Isolation in Italian College Students: The Role of Physical Exercise"

_nutrients, 2020, doi:10.3390/nu12123660_

Round 1

Reviewer 1 Report

I want to thank you for the opportunity to review the article nutrients-1014998 entitled: Dietary habits and psychological states during COVID-19 home isolation: the mediation effect of self-administered physical exercise in Italian college students

The main aim of this longitudinal study seems to be two: 1) to investigate the association between mood state, exercise, and dietary habits, 2) to examine the mediating effects of physical exercise between psychological states and nutrition. According to the authors, this is one of the first attempts to investigate the relationship between nutrition, psychological states and physical exercise using a mediation model.

It is evident that the authors of this manuscript do not write in English as their primary language. While that observation itself is not a limitation, there are many English writing conventions that are not followed throughout the manuscript which distract from the clarity of the sentences, the presentation of the results, etc. The authors should ensure that a native speaker reads the manuscript in order to check for stylistic and grammatical correctness.

I see good potential in this manuscript, although there are some changes that would need attention and revising before being considered for publication.

Title

1.- Please, remove “self-administered” since all measured variables are self-administered.

Abstract

1.- Please, put “physical exercise” instead of “physical activity” throughout the article. The authors have measured exercise in this study not physical activity.

2.- The authors use different names to refer to dietary habits “nutritional choices”, “food choices”, “

 nutritional variables”, “eating behaviours”. Please, put the same name of the variable according to exactly what you have measured.

3.- The aim or aims of this study are not sufficiently clear in the title, abstract, and introduction. In the title the aim is related to the mediating effects of physical exercise between psychological states and nutrition, in the abstract the aim is to investigate the effects of mood states and exercise on nutritional choices, while at the end of the introduction the aim is “to investigate the association between mood state, self-administered exercise, and the dietary habits”. Please, put the different aims of this study.

4.- Please, rewrite this sentence “An observational study was carried out to investigate the effects of mood states and exercise on nutritional choices, on 176 college students during the COVID-19 lockdown: during 21 days, spontaneous nutrition and physical exercise were daily monitored, and the mood states assessed”.

5.- The authors use different names to refer to physical exercise. For example, physical activity or

training load. Please, put the same name of the variable according to exactly what you have measured.

6.- Please, put “longitudinal observational study” in the abstract.

7.- Authors said “Exercise could play a key beneficial role during uncommon situations, such as a home-confinement”. Please, rewrite this conclusion according to your results.

8.- Please, put the mean and standard deviation of the sample.

  1. Introduction

1.- Authors said “…Mood disturbances, such as anxiety, depression, anger and irritability, have been previously reported in patients” Which patients do you refer to?

2.- Authors said “On the flip side of the coin, healthy eating patterns (e.g. 52 Mediterranean diet) are associated with a reduced risk of depression and better mental health”. Please, add a reference to support this claim.

3.- Authors said “Even physical exercise and nutritional choices are strictly interconnected”. Please, see this reference to support this claim.

Geller, K., Lippke, S., & Nigg, C. R. (2017). Future directions of multiple behavior change research. Journal of behavioral medicine, 40(1), 194-202.

4.- Line 69, please put “was reported”.

Prayers in which you put a semicolon are not well understood.

5.- Sentences in which you put a semicolon are not well understood. Por example, “surveys. It has been previously reported that surveys and self-reported food diaries with a short time of data collection are likely to underreporting from participants; the 24-h recall for multiple days has been reported to significantly reduce this effect [13]. Please, rewrite it.

6.- Line 80. Authors said “Exercise has been previously shown to influence both mood states and eating behaviours”. However, the relationship between exercise and mood states is not clearly described in the introduction.

7.- Authors said “In the context of the COVID-19 quarantine, we supposed that the effects of mood disturbances on eating habits could be mitigated by the exercise performed” Has the mediating effects of exercise in the relationship between mood disturbances and eating habits before? I am referring to a different context than the context of the COVID-19 quarantine.

8.- Please, remove “self-administered” since all measured variables are self-administered.

9.- Please, put “people” or “college students” instead of “subjects” throughout the article.

10.- Please, put the main aims of this study.

11.- The introduction should be structured in different sections to make it easier to understand. For example, 1.1. Effect of Exercise on Nutrition, etc.

12.- Perhaps it would be better to contextualise it more in the sample in which the study has been carried out. Could the study sample influence the results? What characterises the sample of university students? Perhaps further studies in this population were lacking during the introduction.

  1. Materials and Methods

2.1. Participants

1.- What type of sampling was used? Convenience?

2.- How was contact made with university students to participate in the study?

3.- What careers did the students who participated in the study belong to?

4.- Much more information seems to be needed about the context in which the study was conducted.

5.- Did university students have online classes?

2.2. Data Collection

1.- Authors said “Before starting to record the diet, subjects were asked to add in a pre-structured form their height and weight, if they were spending the isolation period at home with their families, with friends, or alone, and if they were used to regularly practice sports activity before the COVID-19 epidemics. Were exclusion criteria used?

2.3., 2.4., and 2.5.

1.- It is not clear whether the instruments used to measure diet, physical activity, and psychological states are validated in the university population.

2.- Providing examples in some of the study variables would help to understand them better.

2.6. Statistical Analyses and 3. Results. These sections are fairly well written. Congratulations.

  1. Discussion

1.- Please, see my previous comments regarding the aims of this study.

2.- Authors said “We assumed that these effects are not independent, but exercise might mediate the mood-food relationship, so determining changes in food selection.” Please, add a reference to support this claim.

2.- Authors said “Exercise contributes to a negative energy balance by increasing energy expenditure, but its role on the dietary patterns is not yet fully understood”. This sentence is not in line with the one in the introduction “Even physical exercise and nutritional choices are strictly interconnected”

3.- I do not understand why the following information appears in 4.1. Effect of Exercise on Nutrition

“Wang et al. [9] reported increased consumption of fruits and vegetables among Chinese adults during the COVID-19 lockdown period, particularly in women. The consumption of vegetables and fruits was positively correlated with sleep quality, and consequently with life quality. Furthermore, a diet rich in fresh fruit, vegetables, and fish provides a high anti-inflammatory and antioxidant molecules, including vitamin C, vitamin E, vitamin D, omega-3 and phytochemicals such as carotenoids and polyphenols that could reduce the inflammation and oxidative stress, thereby strengthening the immune system, contributing to the overall health in situations such as the COVID-19 pandemic.”

4.- I do not understand why the following information appears in 4.2. Effect of Mood on Nutrition

Our results are in accordance with Madhav et al. [34], who evidenced that sedentary lifestyle is a significant risk factor of depression among adults; in particular, the authors reported that depression was significantly higher among females.

5.- I suggest you review this article

Lassale, C., Batty, G. D., Baghdadli, A., Jacka, F., Sánchez-Villegas, A., Kivimäki, M., & Akbaraly, T. (2019). Healthy dietary indices and risk of depressive outcomes: a systematic review and meta-analysis of observational studies. Molecular psychiatry, 24(7), 965-986.

6.- I don't understand the title of this section “4.3. Interaction between Mood and Exercise on Nutrition”

What is the objective of this study?

7.- As in the introduction, I suggest that the authors add studies that have examined the objectives addressed in this study in the university population.

Strengths and limitations

1.- Please, add that the convenience sample of this study is one of the main limitations.

2.- Authors said “Moreover, to our best knowledge, this is one of the first attempts to investigate the complicated relationship between nutrition, psychological states and physical exercise using a mediation model.” Is there any study? If there are any previous studies, they should be added in the introduction.

3.- Please, add future avenues of research.

  1. Conclusions

1.- What does mean “spontaneous nutritional choices”? Is that the right name to call that variable?

2.- Please, rewrite this sentence “Overall, exercise led to a healthier lifestyle, and in particular, could play a key role during a situation such as a home-confinement due to a pandemic.” It is not correct to refer a “healthier lifestyle” if only eating habits has been measured.

3.- Please, remove Factor 4 and Factor 6.

Author Response

Dear Reviewer,

Thank you very much for revising our manuscript. We appreciated the comments and suggestions provided, which have helped to improve our manuscript and allowed our review to be presented more clearly.

We have revised the manuscript according to the comments and suggestions and provided a response letter. We have also checked that the manuscript preparation meets all journal preparation guidelines.

We hope that our revisions fulfil the requests of the reviewer and the revised manuscript meets the standards for publication.

Thank you very much for your great help on this matter,

Sincerely yours,
Stefano Amatori

Title

1.- Please, remove “self-administered” since all measured variables are self-administered.

Answer: As suggested “self-administered” was removed from the Title and the text.

Abstract

1.- Please, put “physical exercise” instead of “physical activity” throughout the article. The authors have measured exercise in this study not physical activity.

Answer: As suggested, we changed physical ‘activity’ with ‘exercise’, when referring to our study.

2.- The authors use different names to refer to dietary habits “nutritional choices”, “food choices”, “nutritional variables”, “eating behaviours”. Please, put the same name of the variable according to exactly what you have measured.

Answer: We agree with the reviewer that too many ways for referring to one variable might be confusing. As suggested, we kept “dietary habits” and “nutritional choices” as the only two different ways to call the variable we measured.  We’ve chosen these two terms, for keeping the text readable and not too redundant, as it would be using just one term. We’re confident that now the manuscript is more clear and readable.

3.- The aim or aims of this study are not sufficiently clear in the title, abstract, and introduction. In the title the aim is related to the mediating effects of physical exercise between psychological states and nutrition, in the abstract the aim is to investigate the effects of mood states and exercise on nutritional choices, while at the end of the introduction the aim is “to investigate the association between mood state, self-administered exercise, and the dietary habits”. Please, put the different aims of this study.

Answer: According to the reviewer’s comments the title has been changed in: “Dietary habits and psychological states during COVID-19 home isolation in Italian college students: the role of physical exercise”. Now, the aim of the study has been uniformed throughout the text.

4.- Please, rewrite this sentence “An observational study was carried out to investigate the effects of mood states and exercise on nutritional choices, on 176 college students during the COVID-19 lockdown: during 21 days, spontaneous nutrition and physical exercise were daily monitored, and the mood states assessed”.

Answer: The sentence has been rewritten as suggested by reviewer: “A longitudinal observational study was carried out to investigate the effects of mood states and exercise on nutritional choices, of 176 college students (92 males, 84 females; 23±47 years old) during the COVID-19 lockdown.”.

5.- The authors use different names to refer to physical exercise. For example, physical activity or training load. Please, put the same name of the variable according to exactly what you have measured.

Answer: We agree with the reviewer: the term ‘physical activity’ has been removed throughout the text when we were referring to our study. The term ‘training load’ is present only in the Materials and Methods and the Results sections, as it is the quantitative measure of the exercise that we used for the analyses.

6.- Please, put “longitudinal observational study” in the abstract.

Answer: ‘Longitudinal’ has been added in the abstract, as requested.

7.- Authors said “Exercise could play a key beneficial role during uncommon situations, such as a home-confinement”. Please, rewrite this conclusion according to your results.

Answer: This sentence has been rewritten, as requested.

8.- Please, put the mean and standard deviation of the sample.

Answer: Proportion of males and females, and mean age (± sd) of the sample have been added.

  1. Introduction

1.- Authors said “…Mood disturbances, such as anxiety, depression, anger and irritability, have been previously reported in patients” Which patients do you refer to?

Answer: The word ‘patients’ has been removed since it was inappropriate in this context. Indeed the Ref. [10] (Brooks et al., 2020) is a review that included studies with different populations and quarantine contexts. We hope that now this point is clearer in the text.

2.- Authors said “On the flip side of the coin, healthy eating patterns (e.g. Mediterranean diet) are associated with a reduced risk of depression and better mental health”. Please, add a reference to support this claim.

Answer: A reference [15] has been added, as requested.

3.- Authors said “Even physical exercise and nutritional choices are strictly interconnected”. Please, see this reference to support this claim.

Geller, K., Lippke, S., & Nigg, C. R. (2017). Future directions of multiple behavior change research. Journal of behavioral medicine, 40(1), 194-202.

Answer: We thank the reviewer for this suggestion: it was really interesting and helpful. A citation has been added to support that claim.

4.- Line 69, please put “was reported”.

Prayers in which you put a semicolon are not well understood.

Answer: The semicolon has been removed as suggested, and then the form “was reported” was not necessary anymore. Thanks for the correction.

5.- Sentences in which you put a semicolon are not well understood. Por example, “surveys. It has been previously reported that surveys and self-reported food diaries with a short time of data collection are likely to underreporting from participants; the 24-h recall for multiple days has been reported to significantly reduce this effect [13]. Please, rewrite it.

Answer: The sentence has been changed, as requested. “however” has been added after the semicolon. We hope that now the sentence is clearer.

6.- Line 80. Authors said “Exercise has been previously shown to influence both mood states and eating behaviours”. However, the relationship between exercise and mood states is not clearly described in the introduction.

Answer: We agree with the reviewer on this point. The relationship between mood and exercise has now been described in section 1.3 Mood, Exercise and Nutrition. Despite this, authors think that this relationship does not require a paragraph on its own, because in the study we did not investigate the direct effect of mood on exercise, or vice versa. Our study was focused on the effects of exercise on nutrition and mood on nutrition, and the interaction between these three variables is only taken into account in the mediation model.

7.- Authors said “In the context of the COVID-19 quarantine, we supposed that the effects of mood disturbances on eating habits could be mitigated by the exercise performed” Has the mediating effects of exercise in the relationship between mood disturbances and eating habits before? I am referring to a different context than the context of the COVID-19 quarantine.

Answer: Agreed. Yes, the mediation model has been previously used to investigate the relationship between exercise, mood and nutrition. Some studies in this regard have now been cited in the text. Please see Section 1.3 Mood, Exercise and Nutrition.

8.- Please, remove “self-administered” since all measured variables are self-administered.

Answer: As in the Title, “self-administered” was removed from the text.

9.- Please, put “people” or “college students” instead of “subjects” throughout the article.

Answer: As requested, the word ‘subjects’ have been substituted throughout the article with “participants” or “people” or “college students”, were more appropriate.

10.- Please, put the main aims of this study.

Answer: We thank the reviewer for highlighting this point. The aim of the study has been uniformed throughout the text (in Abstract, Introduction and Discussion sections). It is: “The present longitudinal observational study aimed to investigate the effects of mood states and exercise on nutritional choices of a sample of college students, during the period of forced home-isolation due to COVID-19 outbreak.”.

Furthermore, the sentence “Exercise has been previously shown to influence both mood states and eating behaviours, so we hypothesised that people who would do more physical exercise during the home-confinement period would also have different food choices. In the context of the COVID-19 quarantine, we supposed that the effects of mood disturbances on eating habits could be mitigated by the exercise performed” has been deleted because it was an anticipation of the results.

11.- The introduction should be structured in different sections to make it easier to understand. For example, 1.1. Effect of Exercise on Nutrition, etc.

Answer: Agreed, we added sub-sections in the Introduction, making it easier to understand. The subsections are: 1.1 Mood and Nutrition, 1.2 Exercise and Nutrition, 1.3 Mood, Exercise and Nutrition, 1.4 The COVID-19 context.

12.- Perhaps it would be better to contextualise it more in the sample in which the study has been carried out. Could the study sample influence the results? What characterises the sample of university students? Perhaps further studies in this population were lacking during the introduction.

Answer: As suggested, more information regarding the sample, and the contexts in which the study was carried out, have been added in the Materials & Methods section. Of course, we are aware that our results can only be generalized to the population of Italian university students. However, the results obtained are supported by previous studies and fit into a context of the relationship between mood, exercise and diet, which supports our conclusions. Clearly, we agree with the reviewer that further studies are needed to see whether and how these results can be extended to other strata of the population. Furthermore, a new study conducted on university students has been added in the Introduction (Romero-Blanco et al. 2020).

  1. Materials and Methods 

2.1. Participants

1.- What type of sampling was used? Convenience?

Answer: Yes, convenience sampling was used in this study. This has been now specified in the text.

2.- How was contact made with university students to participate in the study?

Answer: As now reported in the text, the contact was made verbally during the online classes; then, an email was sent to all the students belonging to those classes.

3.- What careers did the students who participated in the study belong to?

Answer: The students were enrolled in three different study careers: pharmacy, biology and sports sciences. This information has been added in the relevant section, in point 2.1.

4.- Much more information seems to be needed about the context in which the study was conducted.

Answer: Some new information regarding the study context has now been added in the text.

5.- Did university students have online classes?

Answer: Yes, in that period students had online classes because the university was closed. As reported in point 3.1, almost all the students spent the lockdown period in their respective homes (with their parents); only seven people stayed in the city where the University is, and lived in a shared house with other students.

2.2. Data Collection

1.- Authors said “Before starting to record the diet, subjects were asked to add in a pre-structured form their height and weight, if they were spending the isolation period at home with their families, with friends, or alone, and if they were used to regularly practice sports activity before the COVID-19 epidemics. Were exclusion criteria used?

Answer: The exclusion criteria have been added at the end of the 2.2 Section, as follows: “Exclusion criteria were: to be on a particular diet (e.g. prescribed by a nutritionist), to suffer from chronic diseases, to be afflicted by COVID-19 before or during, and to have a flu during the data collection period.”

2.3., 2.4., and 2.5.

1.- It is not clear whether the instruments used to measure diet, physical activity, and psychological states are validated in the university population.

Answer: For each instrument, references of previous studies on university population have been reported.

2.- Providing examples in some of the study variables would help to understand them better.

Answer: Regarding exercise, the CR-10 scale has been explicitly described, and more info on its application in training monitoring has been given. Examples of the items comprised in the questionnaires assessing psychological states have been added in their specific sections. 

2.6. Statistical Analyses and 3. Results. These sections are fairly well written. Congratulations.

Answer: Authors are grateful to the reviewer for this comment.

  1. Discussion

1.- Please, see my previous comments regarding the aims of this study.

Answer: As reported in the previous answers, the aim of the study has been uniformed throughout the text.

2.- Authors said “We assumed that these effects are not independent, but exercise might mediate the mood-food relationship, so determining changes in food selection.” Please, add a reference to support this claim.

Answer: As requested, a new reference has been added in the text to support this claim (Annesi et al., 2014).

2.- Authors said “Exercise contributes to a negative energy balance by increasing energy expenditure, but its role on the dietary patterns is not yet fully understood”. This sentence is not in line with the one in the introduction “Even physical exercise and nutritional choices are strictly interconnected”

Answer: This point has been now clarified in the text. Undoubtedly, exercise and nutrition are interrelated. However, the mechanisms which induced changes in nutrition due to increased exercise practice, are not completely understood. Indeed, it is not clear if it is more important the role of exercise in increasing energy expenditure or its action in stimulating psychological changes, then leading to better eating and self-regulatory conducts.

3.- I do not understand why the following information appears in 4.1. Effect of Exercise on Nutrition

“Wang et al. [9] reported increased consumption of fruits and vegetables among Chinese adults during the COVID-19 lockdown period, particularly in women. The consumption of vegetables and fruits was positively correlated with sleep quality, and consequently with life quality. Furthermore, a diet rich in fresh fruit, vegetables, and fish provides a high anti-inflammatory and antioxidant molecules, including vitamin C, vitamin E, vitamin D, omega-3 and phytochemicals such as carotenoids and polyphenols that could reduce the inflammation and oxidative stress, thereby strengthening the immune system, contributing to the overall health in situations such as the COVID-19 pandemic.”

Answer: We agree with the reviewer. As requested, this part has been removed from the text.

4.- I do not understand why the following information appears in 4.2. Effect of Mood on Nutrition

Our results are in accordance with Madhav et al. [34], who evidenced that sedentary lifestyle is a significant risk factor of depression among adults; in particular, the authors reported that depression was significantly higher among females.

Answer: We agree with the reviewer. As requested, this part has been removed from the text.

5.- I suggest you review this article

Lassale, C., Batty, G. D., Baghdadli, A., Jacka, F., Sánchez-Villegas, A., Kivimäki, M., & Akbaraly, T. (2019). Healthy dietary indices and risk of depressive outcomes: a systematic review and meta-analysis of observational studies. Molecular psychiatry, 24(7), 965-986.

Answer: Thanks to the reviewer for suggesting this interesting paper. We use it to improve the Introduction section.

6.- I don't understand the title of this section “4.3. Interaction between Mood and Exercise on Nutrition”

What is the objective of this study?

Answer: We agree with the reviewer that the title of the section was incomprehensible. It has been changed now with a focus on the core topic of the paragraph, which is the mediation effect.

7.- As in the introduction, I suggest that the authors add studies that have examined the objectives addressed in this study in the university population.

Answer: As requested, a new study conducted on a university population has been added in the Discussion section.

Strengths and limitations

1.- Please, add that the convenience sample of this study is one of the main limitations.

Answer: This has been added in the limitation section, as suggested.

2.- Authors said “Moreover, to our best knowledge, this is one of the first attempts to investigate the complicated relationship between nutrition, psychological states and physical exercise using a mediation model.” Is there any study? If there are any previous studies, they should be added in the introduction.

Answer: As suggested, previous studies which used a mediation model to investigate the relationship between mood, exercise and nutrition have been added in the Introduction section. Please, see Section 1.3. Mood, Exercise and Nutrition.

3.- Please, add future avenues of research.

Answer: Some suggestions for future studies have been added in the text, as suggested.

  1. Conclusions

1.- What does mean “spontaneous nutritional choices”? Is that the right name to call that variable?

Answer: Saying that the nutritional choices were “spontaneous” we meant that the participants were not following any specific dietary regimen. However, we recognize that this may create confusion to the reader, so the word ‘spontaneous’ has been removed.

2.- Please, rewrite this sentence “Overall, exercise led to a healthier lifestyle, and in particular, could play a key role during a situation such as a home-confinement due to a pandemic.” It is not correct to refer a “healthier lifestyle” if only eating habits has been measured.

Answer: We agree with the reviewer. The conclusive sentence has been rewritten.

3.- Please, remove Factor 4 and Factor 6.

Answer: Done.

Reviewer 2 Report

Dear Editor
I am pleased to review the manuscript ID: nutrients-1014998, entitled "Dietary habits and psychological states during 2 COVID-19 home isolation: the mediation effect of self-administered physical exercise in Italian college students". The authors attempted to analyze the interplay between dietary habit, mental health and physical exercise during COVID-19 home isolation in Italy college students. They enrolled 176 college students who were socially isolated by COVID-19 lockdown. During 21-day period, dietary record and physical exercise were monitored, and the mood states were assessed. They found exercise load was positively associated with fruit, vegetables and fish consumption. Depression and quality of life were inversely associated with cereals, legumes and low-fat meat intake  Exercise could mediate the effect of mood states on fruit, vegetables and fish consumption. In overall, the study was novel and interesting, and the article was well presented and informative. Some concerns, however, needs clarification.

  1. The definition of home-isolation is not really clear. Would they be able to go outside for physical activity? Would they allow to eat out or limit to eat at home? The key point is the availability and accessibility of food resource, which may influence what they eat.
  2. About exercise monitoring, only structured exercise sessions were considered. What does “ structured exercise session”? May explain it in detail.
  3. Were the participants and/or their family/friends afflicted with COVID-19 or have flu symptoms during this home confinement? The symptoms may affect appetite, which may affect food choice.
  4. Dietary habit and food choice somewhat depend on personal preference before start of the study. Someone inherently favors meat more than vegetables, and the vice versa. How could this confounder be avoided or considered in this analysis.
  5. Thirty-percent participants could not be able to accomplish the study. The participants who completed the study might have better mental status than those who did not. It is better to explain why they drop out.

Author Response

Dear Reviewer,

Thank you very much for revising our manuscript. We appreciated the comments and suggestions provided, which have helped to improve our manuscript and allowed our review to be presented more clearly.

We have revised the manuscript according to the comments and suggestions and provided a response letter. We have also checked that the manuscript preparation meets all journal preparation guidelines.

We hope that our revisions fulfil the requests of the reviewer and the revised manuscript meets the standards for publication.

Thank you very much for your great help on this matter,

Sincerely yours,
Stefano Amatori

1. The definition of home-isolation is not really clear. Would they be able to go outside for physical activity? Would they allow to eat out or limit to eat at home? The key point is the availability and accessibility of food resource, which may influence what they eat.

Answer: More detailed information regarding the context in which the study was conducted has been added in the Methods section. People were not allowed to go out for physical activities, and restaurants were closed (so they were forced to eat at home). However, food availability and accessibility were not a problem at any time.

2. About exercise monitoring, only structured exercise sessions were considered. What does “structured exercise session”? May explain it in detail.

Answer: As people were not allowed to go outside for exercise, some of them practised training sessions indoors. For each day, they were asked to answer yes/no to “Did you train today?”. Exercise sessions mainly included free-weight exercises, runs or rides on indoor trainers, rope jumping. For each training session, they indicated on the dairy the duration in minutes (ex. 55 min), and the perceived effort of the whole session using the modified CR-10 scale. This part has been now detailed in the relevant section.

3. Were the participants and/or their family/friends afflicted with COVID-19 or have flu symptoms during this home confinement? The symptoms may affect appetite, which may affect food choice.

Answer: None of the participants was afflicted by COVID-19, before or during the study, and neither had flu in that period of time. This has been added in the exclusion criteria section. However, we can just say that none of them showed symptoms related to COVID-19, but we cannot know if they were asymptomatic, and neither we or them knew that. Unfortunately, we are not aware whether their family members or friends were afflicted by COVID-19 or flu in that period.

4. Dietary habit and food choice somewhat depend on personal preference before start of the study. Someone inherently favors meat more than vegetables, and the vice versa. How could this confounder be avoided or considered in this analysis.

Answer: We agree with the reviewer, saying that people may have different eating habits. In order to keep this issue under control, the GLM analysis, as reported in section 2.6 (Statistical Analyses), considers the ID (subject’s identifier) of each subject as a random factor. Therefore, the results of multivariate analyses take into account the eating habits of the subjects, and the results obtained are net of the latter.

5. Thirty-percent participants could not be able to accomplish the study. The participants who completed the study might have better mental status than those who did not. It is better to explain why they drop out.

Answer: Thanks for highlighting this point. Those participants (29.6%) did not drop out (i.e. that they did not abandon the study), but simply they were non-responders. They were invited to take part in the study, but they chose to not participate.

Round 2

Reviewer 1 Report

I want to thank you for the opportunity to review again the Manuscript ID: nutrients-1014998, titled Dietary habits and psychological states during COVID-19 home isolation in Italian college students: the role of physical exercise for Nutrients.

The main aim of this longitudinal study seems to be two: 1) to investigate the association between mood state, exercise, and dietary habits, 2) to examine the mediating effects of physical exercise between psychological states and nutrition. According to the authors, this is one of the first attempts to investigate the relationship between nutrition, psychological states and physical exercise using a mediation model.

I was surprised by the quality of the authors’ revision effort. Frankly, I thought the reviewers were asking too much of the authors to handle/revise, but the author team carried out an excellent revision. Specifically, the authors have strengthened the introduction and discussion section. My personal recommendation is to accept the manuscript without changes. However, I don't feel qualified to judge about the English language and style. Perhaps extensive editing of English language and style should be required.